# Treatment of Y-T Humeral Fractures with Polyaxial Locking Plate System (PAX) in 14 Dogs

**DOI:** 10.3390/vetsci9070310

**Published:** 2022-06-22

**Authors:** Filippo Maria Martini, Paolo Boschi, Filippo Lusetti, Chadi Eid, Andrea Bonardi

**Affiliations:** 1Department of Veterinary Science, University of Parma, Via del Taglio 10, 43126 Parma, Italy; filippomaria.martini@unipr.it; 2Ortovet stp s.r.l., Piazza Alessandrini 2/D, 43036 Fidenza, Italy; filippo.lusetti@ortovet.org (F.L.); chadi.eid@ortovet.org (C.E.); andrea.bonardi@ortovet.org (A.B.)

**Keywords:** polyaxial locking plate, internal fixation, bicondylar fracture, dog, humerus

## Abstract

The aim of this study is to report the results and to review the outcome of 14 cases of Y-T humeral fractures repair using paired polyaxial locking system (PAX) plates through a combined medial and lateral approach. Fourteen consecutive dogs, with traumatic humeral Y-T fractures, met the inclusion criteria. This study includes signalment, preoperative radiographs, type of implants, radiographic bone healing assessment, complications, range of motion (ROM) of the elbow and limb function evaluated at 120 days after surgery. Postoperative radiographs revealed adequate anatomic reconstruction, and in all cases, bone healing has been achieved. No implant failure was observed. Functional outcome was excellent in 7 dogs (no lameness and preserved ROM), good in 4 (slight lameness and moderate ROM reduction) and discrete in 2 (mild lameness and severe ROM reduction). Complications were encountered in 2/14 patients with implant-associated infection resolved after long-term antibiotic treatment and implant removal. The PAX system is shown to be a valid alternative for the treatment of Y-T humeral fractures, offering the benefit of polyaxial insertion of locking screws. The possibility of angle locking screws is helpful in the distal humeral bicondylar fractures, providing additional options for screw placement in juxtarticular fractures, avoiding fracture lines or other implants.

## 1. Introduction

Fractures of the humerus represent 8 to 12% of all fractures in dogs, with the humeral condyle the most commonly affected region [1]. The Y-T humeral fractures, also known as distal humeral bicondylar fractures [2] may be simple (3-fragments) or comminuted in the supracondylar region [3] and they represent a unique therapeutic challenge [4,5]. As with all articular fractures, the aim of management should include rigid fracture fragment fixation and precise reconstruction of the articular surface [4,5,6]. Several approaches for the treatment of Y-T humeral fractures have been described as ulnar osteotomy, triceps tenotomy, and combined medial and lateral approach to the distal humerus [4,7]

Considering the internal fixation, the use of locking implants for humeral fracture repair has already been described [2,8,9]. The benefits of locking plates include maintenance of periosteal vascular integrity, decreased risk of loss of fracture reduction during screw tightening, a minimized need for precise anatomic plate contouring and reduced screw number for each bone segment [10]. Traditionally, one of the major limitations in the use of locking plates is the inability to angle the screw in the plate hole [11] because a proper screw head-plate coupling, is achieved by inserting the screw into the plate at a precise perpendicular angle [12].

The polyaxial advanced locking system (PAX; Securos Surgical AmerisourceBergen, Fiskdale, MA, USA) is an implant made of titanium alloy (Ti6Al4V) that should determine theoretically minimal tissue reaction with a potentially lower infection incidence [13]. The PAX system was designed with the particular benefits of locking plate technology while offering a polyaxial insertion option of the screws, which can be angled multidirectionally. The PAX-system’s screw locking mechanism is achieved as the sharp threads of the harder screw head engages and “cut” threads within the holes of the plate during insertion. As in other polyaxial implants, the best locking mechanism is achieved when the screw is inserted perpendicular to the hole of the plate, but the multidirectional screw insertion with an angulation up to 10° within the plate in the Pax System allows an appropriate locking strength [11,12,14,15]. The possibility to angulate the screw is particularly advantageous when screws need to be directed away from articular surfaces, fracture lines or when challenging fracture site exposure would have made a perpendicular screw insertion difficult [12].

The distal humeral anatomy is characterized by the presence of the supratrochlear foramen, the thin lateral epicondylar ridge, the irregular surfaces of the metaphysis and the proximity to the elbow all resulting in minimal area for safe and effective implant placement [16]. The features of the PAX system allow for proper stability and fixation in this anatomical region.

The purpose of this paper is to describe the repair of Y-T humeral condylar fractures using two PAX plates applied through a combined medial and lateral surgical approach and to report the outcome and complications in 14 dogs.

## 2. Materials and Methods

### 2.1. Inclusion Criteria

Between January 2016 and August 2019, 21 patients were treated for the repair of humeral condylar Y-T-type fractures. The patients included in this study underwent Y-T humeral fractures repair using paired PAX plates through a combined medial and lateral approach and transcondylar screw placement. Fourteen client-owned dogs met the inclusion criteria. For each patient informed consent was obtained. Every dog underwent pre and post-operative and short term radiographs and clinical examination. Moreover, we collected signalment, affected limb with time of bone healing, complications, osteoarthritis (OA) evaluation, range of motion (ROM) of the elbow, degree of lameness and outcome have been indicated, except for one dog who died 15 days after surgery.

Complications were classified as minor, major and catastrophic; minor complications did not require additional surgical or medical treatment to resolve, while major complication required surgical revision under anesthesia. Catastrophic complications have been defined as those that result from an unacceptable permanent function directly related to death or is cause for euthanasia [17] Table 1.

### 2.2. Clinical and Radiographic Assessment

Before surgical treatment, each patient underwent standard orthogonal radiographic projections of the humerus of the affected and contralateral limbs. The contralateral humerus was also radiographically investigated to evaluate the presence of incomplete ossification of the humeral intracondylar fissure (HIF) as it is a predisposing condition for Y-T fractures [18]. Postoperative radiographs examination was performed in both orthogonal standard projections of the humerus to evaluate the reduction of the fracture and the position of the implants. Bone healing, decreed through callus formation, fracture line disappearance and stage of union on day 30, 60 or 90 from surgical treatment, has been established by observing the two standard radiographic views of the humerus, obtained under sedation [19]. Clinical and radiographic follow-up (FU) of each dog was performed at 30 days after surgery (FU 30), except for one patient who died previously (case 13). Additional clinical and radiographic examination 30 days later (FU 60) or 60 days later (FU 90) were recommended to the patients with no evidence of bone union based on the callus stage. At FU 120, all subjects, except for one patient who died previously (case 13), were evaluated, and each patient underwent sedation [20] and radiographic study was performed to assess implant position, necessity of implant removal and osteoarthritis development. OA was graded: absent or minimal evidence of OA with no radiographic signs; mild with osteophytes smaller than 2 mm, moderate with osteophytes between 2 and 3 mm and severe osteoarthritic changes with osteophytes larger than 5 mm [21,22]. Limb functionality was assessed on day 120 by lameness scoring on the following scale: of 5:0, absence of lameness; 1, slight lameness, impaired movement with slightly altered limb load even if limited; 2, mild lameness impaired movement with maintained functionality; 3, moderate lameness, impaired movement and impaired functionality, sometimes with lack of limb support; 4, altered movement with lack of limb support [21]. ROM of the elbow was evaluated on day 120 and classified in three degrees: preserved (absence of alterations), moderate reduction (reduction from 20–40°) or severe reduction (reduction > 40°) [23]. Considering the overall radiographic and clinical assessment, the presence or not of post-operative complications and the development of OA, the final outcomes were classified as excellent, good, discrete and poor. Radiographic study was performed to assess implant position, necessity of implants removal and osteoarthritis development.

### 2.3. Surgical Technique

All patients underwent the same anesthetic protocol following guidelines of WSAVA Global Pain Treatise: premedication with combination of methadone (0.2 mg/kg) and dexmedetomidine (3 mcg/kg) intramuscularly, induction with propofol to effect until endotracheal intubation was achievable and all dogs were administered isoflurane mixed in a carrier gas of 100% oxygen. Fentanyl was used as intraoperative pain therapy with the initial dose of 5 mcg/kg/hr, and adjustments were determined based on fluctuations in physiological parameters (e.g., ±20% change in heart rate and blood pressure) and anesthetic depth. After induction of general anesthesia [20], each patient was positioned in dorso-lateral recumbency with the affected limb free to be approached both medially and laterally and the contralateral limb extended caudally. All dogs received antibiotic treatment with cefazolin 20 mg/kg administered IV at the time of induction of general anesthesia and then every 90 min until the end of the procedure. In all patients, a medial approach to the humerus was first performed to treat the medial condylar fracture with PAX plate in order to transform the bicondylar fracture in a lateral monocondylar fracture. In some cases, it was useful to apply supplementary implants, such as one or more supracondylar position or lag cortical screws or cerclage wire to facilitate anatomical reconstruction. Subsequently, a lateral approach was performed to reduce and fix the lateral condylar fracture through the application of a transcondylar cortical stainless steel screw using the “inside-out” or “outside-in” techniques with the screw placed in position or lag fashion according to surgeon’s preference [7,8]. Finally, a lateral PAX plate was applied. Both plates were applied respectively on the effective medial and lateral side of the bone, perpendicularly to the frontal plane, regardless of the screws’ orientation needed to achieve fixation. Surgical approaches were closed routinely.

### 2.4. Postoperative Care

Nonsteroidal anti-inflammatory drugs (Carpofren 3 mg/kg) were administered for 7–14 days postoperatively, while antibiotic therapy with Cefadroxil (20 mg/kg orally twice daily) was prescribed for 1 week postoperatively in all dogs. After the surgery, the owners were instructed to perform passive flexion-extension movements of the elbow and the carpus three times a day for 5 min starting two weeks after surgery [24]. Restriction of physical activity by cage rest is also recommended until radiographic evidence of bone healing.

## 3. Results

Fourteen dogs were included in the study: French Bulldog (3), English Pointer (2), Mixed breed (2), Labrador Retriever (1), English Springer Spaniel (1), English Setter (1), German Pointer (1), Whippet (1), Toy Poodle (1) and Lagotto Romagnolo (1). The age of the dogs ranged from 8 months to 7.3 years (median 4.1 years), the weight ranged from 4.2 kg to 38.5 kg (mean 17.7 kg). Three (21.5%) dogs (cases 8,10 and14) were considered skeletally immature, with distal humeral physes still visible on radiographs of contralateral limb at the time of presentation [9]. None of the cases included in this study had radiographic evidence of HIF on the contralateral limb (Table 1).

### 3.1. Post-Operative Radiographic Assessment

All the fractures (14/14) were stabilized by the use of a double PAX plate (RP = locking reconstruction plates and/or SP = locking limited contact straight plates) and transcondylar cortical screw: medial SP 2.0 plate and lateral RP 2.0 plate in one dog weighing 4.2 kg (case 12); double RP 2.4 plate in one dog weighing 7.3 kg (case 8); double RP 2.7 plate in 4 dogs (cases 6-7-10-14) and medial SP 2.7 plate and lateral RP 2.7 plate in one dog (case 4) in patients between 9–14.5 kg; medial RP 3.5 plate and lateral RP 2.7 plate in 3 patients (cases 2-3-13) and medial SP 3.5 plate and lateral RP 2.7 plate in one dog (case 5) between 16.7–23 kg; double RP 3.5 plate in one dog weighing 25 kg (case 9); medial SP 3.5 plate and lateral RP 3.5 plate in two dogs (cases 1-11) between 28–38.5 kg. In seven dogs, 2.7 mm transcondylar cortical screws were applied; in six cases, 3.5 mm transcondylar cortical screws were used, and in one case, 4.5 mm transcondylar cortical screws were used. Additional supracondylar implants were applied in 7/14 cases, including cerclage wire (n = 1), position cortical screw (n = 4), lag cortical screw (n = 1) and Kirschner wires (n = 2) (Table 2). In the present series, a total of 118 screws were applied in 28 PAX plates; in particular, we used 61 screws proximal to the fracture with 36 screws positioned medially (range 2 to 4 and median value 2) and 25 screws were positioned laterally (range 1 to 2 and median value 2); 57 screws were inserted distal to the fracture with 29 screws positioned medially (range 2 to 3 and median value 2) and 28 screws positioned laterally (range 1 to 2 and median value 2). The median value of plate-screw density was found to be 0.57 calculated for both the lateral (range 0.40 to 0.80) and medial (range 0.44 to 0.87) sides. The post-operative radiographic study showed in each clinical case an adequate reduction of the fracture, without evidence of discontinuity of the articular surface and the alignment was adequate in all the cases performed. Although no intraoperative imaging supports (i.e., Fluoroscopy or intraoperative x-ray) were used, no locking screws were inserted in the joint space or within the fracture line. Bone healing assessed radiographically was obtained in all patients, except for case 13; the patient died fifteen days after surgery for causes not related to the surgical treatment. In 3/14 dogs, bone union was evident at the first post-operative FU at 30 days (Figure 1); in 8/14 dogs, the bone healing occurred between the first and the second postoperative FU; therefore, these fractures were considered healed at FU 60; in 2/14 patients, due to implant associated infection, bone healing was detected during the third FU at 90 days postoperatively. Radiographic study was performed at FU 120 to observe implant position and possible osteoarthritis development. In no case, implant failure, breakage or loosening of the screws occurred. OA was scored as follows: absent (6/14), mild (5/14) and moderate (2/14) and severe (0/14) (Table 1).

### 3.2. Complications

Major septic complications were observed in 2/14 patients, respectively 4 (case 5), and 2 (case 3) weeks post-operative. In case 5, the dog presented subcutaneous swelling and draining tract on the lateral side of the elbow. Implant-associated infection based on percutaneous positive bacteria culture (*Staphylococcus* spp.) was diagnosed at FU 30. Susceptible antibiotic therapy (amoxicillin/clavulanic acid was prescribed until bone healing), and afterward implant removal was necessary. In the other case (n = 3), the patient had surgical wound dehiscence on the lateral humeral side 2 weeks post-surgery and healed by secondary intention. Bacteria culture was performed and Staphylococcus spp. was isolated. Amoxicillin/clavulanic acid has been administered and protracted until implant removal. The infections were managed based on antibiogram results and antibiotic therapy until complete bone healing. After establishing successful bone union, at FU 90, implant removal was necessary. In both cases, the lateral plate and the transcondylar screw were removed to resolve the infection. Despite the underlying infection, in both cases, no instability or loosening of the implants was detected during their removal.

### 3.3. Clinical Assessment

At FU 120, seven patients had no lameness and showed normal ROM of the elbow joint; 4 dogs presented good functionality of the limb with first-grade lameness and moderately reduced joint ROM; and two dogs presented second-grade lameness with the elbow joint ROM being severely reduced with a tendency to worsen after intense physical activity [23].

### 3.4. Outcomes

Considering both clinical and radiographic results obtained at FU 120, we did not report a poor overall outcome in any case. The overall outcome was considered excellent in 7 dogs (cases 4-7-8-10-11-14), good in 4 dogs (cases 1-2-6-9) and discrete in 2 dogs (cases 3–5) (Table 1).

## 4. Discussion

The general outcome using PAX in this case series compared favorably with previously published reviews. In 2 cases, we reported major complications that required implant removal once the bone was healed. No radiographic signs of HIF as predisposing factor to Y-T fractures were found; however, in 3/14 dogs (cases 2-3-9), the surgeon had a strong intra-operative suspicion of a pathological fracture, with an underlying HIF, given the severe bone sclerosis involving the intercondylar fractures [25]. Each case has been successfully treated through a bilateral surgical approach, obtaining an excellent exposure of the fracture, anatomic reconstruction of the articular surface and rigid internal fixation [3]. Our study is not dissimilar to other repair strategies of Y-T fractures already described with locking implants [2,8]. This article seems to report results better than those of the study by Mckee et al. about the reduction of the condylar (articular) fracture that is a pre-requisite for obtaining a satisfactory outcome [2]. To our knowledge, in veterinary surgery, the results regarding the clinical use of polyaxial locking plate are only reported with the use of the PAX system, which showed good clinical outcomes, and in human medicine, biomechanical and clinical studies showed general good results with the use of polyaxial locking plates [12,26]. The implant choice in our cases was based on the fracture’s configuration, the weight of the dog and its temperament. PAX offers the possibility to choose between SP and RP implants; both plates have the same thickness, but PAX SP provides a significantly larger section modulus than PAX RP, which makes the plate stronger [11]. In case of bone loss in the supratrochlear region we preferred to use PAX SP for its properties described above. In all the cases where we used the SP plate (cases 1-4-5-12), it was applied medially which is the tension side of the distal humerus [5]. Additionally, the straighter shape of the medial epicondylar crest requires minimal plate contouring and allows for more bone stock to be available for screw purchase [3,5,27]. In ten cases, we used RP plates both on the medial and lateral side of the humerus, and in cases 2, 3 and13, because of the anatomical features of the bone or due to fracture configuration, we combined different RP implant sizes between the lateral and the medial aspects of the bone, using the larger plate on the medial side and the smaller on the lateral side (Table 2).

Plating the distal humerus sometimes requires extreme contouring of the plate [8]. In human medicine, the use of anatomical pre-contoured locking implants seems to be very useful in order to increase the stability of the construct and reduce soft tissue irritation and surgical time [28]. The contouring of locking plates is not strictly necessary; nevertheless, we found it very useful to contour the plates to better address the bone in order to limit the juxtarticular soft tissue impingement. The proper shaping of the PAX RP is achieved using the PAX reconstruction bending plier. This device allows the plate to be bent in multiple planes, while preventing polyaxial plate holes from deformation [14]. The bending plier is necessary in order to avoid multiple bending corrections in the attempt to reproduce the complex footprint of the distal humerus, as repetitive bending may lead to metal fatigue in the contoured areas, increasing the risk of premature plate failure [29]. The plate contouring reported in previous studies seems to condition the direction of the locking screws, while the direction of the PAX system screw is less influenced by the plate contouring. Garcia et al. reported the application of medial epicondylar plates on the medial aspect of the humerus, while the lateral epicondylar plates were applied on the lateral aspect in 12 fractures and caudally in 15 fractures [9]. We hypothesized that the authors, not being able to direct the locking screws using the locking compression plate system (LCP), were forced to apply the plate in such a way to avoid invading the fracture lines or joint space. In our cases, the parallel application of the plates on true medial and lateral bone sides has been made possible by the use of PAX system (Figure 2).

Most of the time, insertion of bicortical locking screws was possible. In our cases, implant failure was not observed. This satisfactory result may be due to the plate’s position on the medial and lateral aspects of the humerus. In the present series, a total of 28 PAX plates and 118 PAX screws were placed, and no screw breakage, loosening or pull-out were observed. Two recent papers reported the treatment of Y-T fractures with the use of LCP [8,9]. In both studies, a combination of locking and non-locking screws were used. Moffat et al. used 4 and 3 screws (median value) for proximal/distal bone fragment medially and laterally, respectively. Garcia et al. did not report a precise number of screws used, but relying on the observation of the published figures and tables, we hypothesize that they used a greater number of screws per case compared to our case series. In fact, we report a median value of only 2 screws for proximal/distal and medial/lateral bone segment without observing failures. We speculate that the greater number of screws used in Moffat and Garcia’s studies compared to our case series may be due to the combination of locking and non-locking screws. Ness, Moffat et al. and Garcia et al. [2,8,9] did not indicate the plate-screw density. In locking plates applied in diaphyseal fractures, values of plate screw density below 0.5 to 0.4 has been recommended, indicating that less than half of the plate holes are occupied by screws, but it is unknown if these recommendations should be applied in articular fractures treated with double plating [30]. In our case series, the plate screw density was 0.50 (median value) on the medial side and 0.57 on the lateral side. A lower screw density could have been obtained by using longer plates with the risk of reducing the construct’s stiffness, which is a prerequisite for articular fractures treatment. A higher screw density would have resulted in contrast with the general screw density indications for locking plate application. Thus, in order to obtain a screw density as close as possible to that indicated by Gautier et al. [30] for diaphyseal fractures we preferred the use of shorter plates. Future studies would be interesting to define the minimum number of screws needed and the biomechanical properties of the construct considering orthogonal or parallel plating. In some cases, additional implants, such as cerclage wires, Kirschner wires and/or position or lag screws were used to help reduce the fracture before plate fixation [2]. The possibility to use a polyaxial locking screw allowed to avoid fissures, joint space, and other implants and to increase bone purchase [26]. These features make the PAX system very versatile and, compared to fixed-angle locking plates, it may be advantageous to use polyaxial locking plates for the treatment of distal humerus fractures.

The elbow joint ROM was evaluated in all dogs included in this series as good in 11/14 cases; in particular, 7 patients maintained normal ROM and 2 patients had a severe ROM reduction. In human medicine, prolonged immobilization of the elbow predisposes to joint stiffness, muscle atrophy and permanent functional impairment [31]. Recommendations and protocols for rehabilitation after elbow surgery are also described in veterinary medicine. Reestablishing full elbow extension is the primary goal of early ROM activities to minimize the occurrence of elbow flexion contractures [24].

In 3 skeletally immature dogs of this series, bone union was evident at the first post-operative check at FU 30. The quick process of bone healing in these cases could be associated with favorable biological factors of immature animals that may play a potential role in success [9]. Garcia et al. reported bone healing of some skeletally immature patients to be observed 6 weeks after surgery. The authors performed the first postoperative radiographic study at that time. It is possible to hypothesize that bone healing may have occurred 30 days postoperatively as in our report, a couple of weeks prior to the authors’ established follow-up date.

In 2/14 cases, a major complication represented by implant-associated infection was observed. Implant-associated infection resolution generally requires complete implant removal [32] In boh dogs, based on culture and susceptibility results, amoxicillin/clavulanic acid use promoted the bone union. In both patients, since we had the strong clinical suspicion that the infection was localized on the lateral side, we decided on a progressive implant removal starting from the lateral side (plate and transcondylar screw). The decision for a progressive implant removal was made to gradually destabilize the underlying bone in an attempt to avoid re-fracture. In both cases, after the lateral implant removal, the dogs, at FU 120, did not show any need for medial implant removal. In both subjects, delayed bone healing and mild osteoarthritis development [20] was observed. Both dogs presented mild limb function with severe reduction in elbow flexion. Nevertheless, these patients were able to exercise without restriction, but intensive activity caused lameness that resolved with nonsteroidal anti-inflammatory medication. The frequency of post-traumatic osteoarthritis in the dogs after humeral condylar fracture repair is unknown, and accuracy of articular fracture reduction did not correlate with follow-up osteoarthritis score [3,22]. Short-term outcome [16] and the small number of clinical cases, which are the main limitations of our study, do not allow to precisely define the development of post traumatic OA. It is probable that OA developed and progressed in all cases [22]. Another limit of this study is the retrospective nature of the work and the post-operative subjective evaluation. It is useful to remember that all previous Y-T fractures studies with locking implants based their results on subjective assessments of the gait functionality [33] following repair of the fracture.

## 5. Conclusions

In all the treated cases, a correct anatomical reduction was obtained with rigid double PAX plates fixation achieved through a combined medial and lateral approach. No implant failure or screw loosening were observed. The functional outcome following this surgery was excellent in 7 cases, good in 4 dogs and discrete in 2 cases in which implant removal was required due to infections. The general outcome of our case series compared favorably with previous reports. The use of the PAX system, taking advantage from polyaxial locking screws, can be a valid alternative to other implants previously reported for the treatment of Y-T humeral fractures in dogs.

## Figures and Tables

**Figure 1 vetsci-09-00310-f001:**
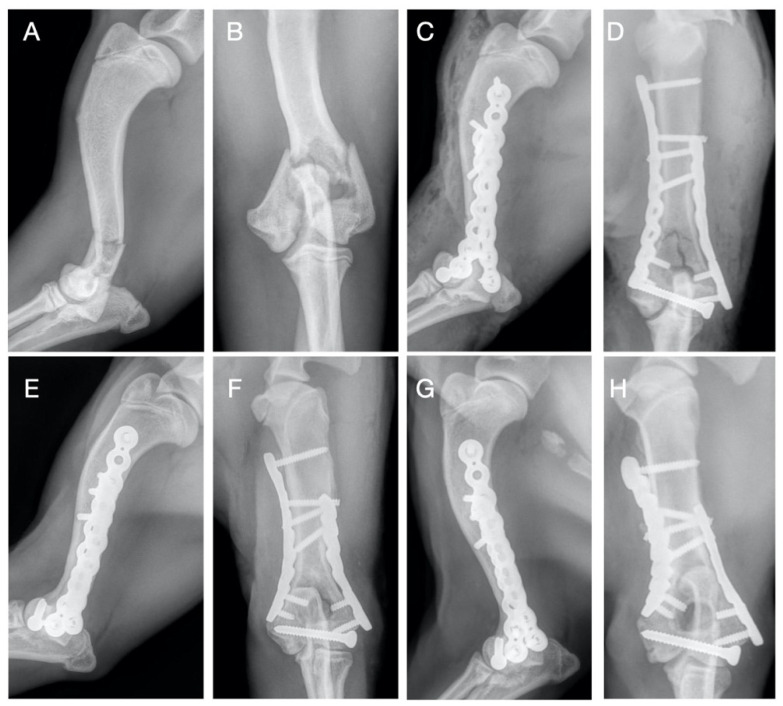
Case 8. Preoperative (**A**,**B**) craniocaudal and medio-lateral projection showing simple distal humeral bicondylar fracture. Immediate postoperative (**C**,**D**) medio-lateral and caudocranial views showing a double 2,4 mm RP PAX plate and 2,7 mm transcondylar lag screw. Medio-lateral and caudocranial projections at 30 days (**E**,**F**) showing bone union at the diaphyseal fracture and primary reduction of the condylar fracture; follow-up 120-day radiographic evaluation (**G**,**H**) with medio-lateral and craniocaudal projections showing no implant failure.

**Figure 2 vetsci-09-00310-f002:**
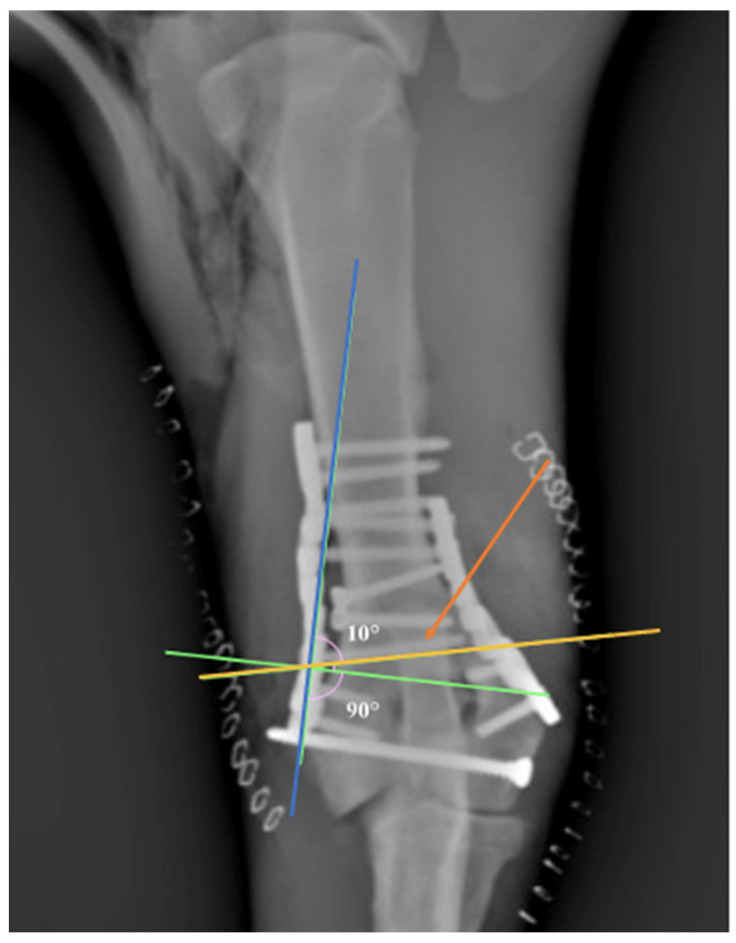
Note the angulation (10°) of the screw (orange arrow) in an attempt to avoid the joint space. Note that the screw inserted at 90° could invade the joint space (green line). Yellow line: screw direction. Blue line: orientation of the plate’s hole.

**Table 1 vetsci-09-00310-t001:** Signalment, fracture healing, complications, lameness and ROM flexion classification at follow-up 120 day.

Case	Signalment	Limb Affected	Time to Fracture Healing (days)	Complications	Score OA FU 120	Lameness Classification FU 120	ROM Flexion Classification FU 120	Overall Outcome FU 120
1	Labrador Retriever M, 1 YO 6 MO 38.5 Kg	R	60	None	Mild	I°	Moderate reduction	Good
2	German Pointer F, 4 YO, 23 KG	L	30	None	Mild	I°	Moderate reduction	Good
3	English Pointer F, 4 YO, 22.5 Kg	L	90	Major: implant associated infection resolved with antibiotic treatment and implant removal	Moderate	II°	Severe reduction	Discrete
4	Whippet F, 4 YO 2 MO. 14.5 Kg	L	60	None	Absent	Absent	Conservate	Excellent
5	Mixed M, 3 YO 8 MO, 21 Kg	L	90	Major: implant associated infection resolved with antibiotic treatment and implant removal	Moderate	II°	Severe reduction	Discrete
6	Lagotto Romagnolo M, 4 YO 6 MO, 14 Kg	L	30	None	Mild	I°	Moderate reduction	Good
7	English Springer Spaniel F, 4 YO 6 MO 14.8 Kg	L	30	None	Absent	Absent	Conservate	Excellent
8	French Bulldog M, 9 MO, 7.3 Kg	L	30	None	Absent	Absent	Conservate	Excellent
9	English Pointer F, 7 YO 3 MO, 25 Kg	L	60	None	Mild	I°	Moderate reduction	Good
10	French Bulldog M, 8 MO, 9 Kg	R	30	None	Absent	Absent	Conservate	Excellent
11	Mixed M, 5 Y 4 MO, 28 Kg	L	60	None	Absent	Absent	Conservate	Excellent
12	Toy Poodle F, 1 YO 4 MO,4.2 Kg	L	30	None	Absent	Absent	Conservate	Excellent
13	English Setter M, 7 YO, 16.7 Kg	R	Dead before first check	-	-	-	-	-
14	French Bulldog M, 10 MO, 9.7 Kg	L	30	None	Mild	Absent	Conservate	Excellent

YO, years old. MO, months old. M, male. F, female. R, right. L, left. FU 120, follow-up 120 day. OA, osteoarthritis.

**Table 2 vetsci-09-00310-t002:** Medial and lateral implants.

Case	Side	Plate	Screws			Plate Screw Density	Transcondylar lag Screw (mm)	Additional Implants
			Proximal to Fracture	Distal to Fracture	Total			
1	Medial	SP 3.5 9H	3	2	4	0.55	3.5	Position cortical screw
	Lateral	RP 3.5 7H	2	2	4	0.57		
2	Medial	RP 3.5 8H	4	3	7	0.87	4.5	/
	Lateral	RP 2.7 7H	2	3	5	0.7		
3	Medial	RP 3.5 7H	3	2	5	0.7	3.5	/
	Lateral	RP 2.7 7H	2	2	4	0.57		
4	Medial	SP 2.7 9H	3	2	5	0.55	2.7	Antirotational Kirschner
	Lateral	RP 2.7 7H	2	2	4	0.57		
5	Medial	SP 3.5 9H	2	2	4	0.44	3.5	Antirotational Kirschner and lag cortical screw
	Lateral	RP 2.7 9H	2	2	4	0.44		
6	Medial	RP 2.7 8H	2	2	4	0.5	2.7	Cerclage wire
	Lateral	RP 2.7 8H	2	2	4	0.5		
7	Medial	RP 2.7 7H	2	2	4	0.57	2.7	/
	Lateral	RP 2.7 6H	2	2	4	0.66		
8	Medial	RP 2.4 9H	2	2	4	0.44	2.7	/
	Lateral	RP 2.4 7H	2	2	4	0.57		
9	Medial	RP 3.5 8H	2	2	4	0.5	3.5	Position cortical screw
	Lateral	RP 3.5 6H	2	2	4	0.66		
10	Medial	RP 2.7 7H	2	2	4	0.57	2.7	Position cortal screw
	Lateral	RP 2.7 5H	1	1	2	0.4		
11	Medial	SP 3.5 9H	3	2	5	0.55	3.5	Position cortical screw
	Lateral	RP 3.5 7H	2	2	4	0.57		
12	Medial	SP 2.0 7H	2	2	4	0.57	2.7	/
	Lateral	RP 2.0 6H	2	2	4	0.66		
13	Medial	RP 3.5 7H	2	2	4	0.57	3.5	/
	Lateral	RP 2.7 7H	2	2	4	0.57		
14	Medial	RP 2.7 7H	2	2	4	0.57	2.7	/
	Lateral	RP 2.7 5H	2	2	4	0.8		

H, holes. RP, recostruction plate. SP, straight plate.

## Data Availability

The data present in this study are available within the article.

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
