# Peer review of "Treatment of Y-T Humeral Fractures with Polyaxial Locking Plate System (PAX) in 14 Dogs"

_vetsci, 2022, doi:10.3390/vetsci9070310_

Round 1
Reviewer 1 Report
The author's presented a retrospective of 14 cases that used a specific implant design and report on the results of those interventions.
Please see comment below for suggestions to improve the manuscript's readability and usefulness for readers. Mainly I think more direct comparisons to veterinary literature in regards to their outcome to those of previous published results would be helpful and well as improved images depicting the polyaxial capability in use - possibly intraoperative image of the screw in the plate showing angled insertion or better radiographic images - or annotation of images to better demonstrate angle insertion of screws.
Line 46: change peculiar to particular
Line 70: change are to were
Line 77: wording awkward: possibly change to “the each patient underwent standard orthogonal radiographic projections of the humerus of the affected and contralateral limbs.”
Line 84: change throught to through
Line 83-86 – run on sentence – reword please
Line 87 – why not just say radiographic follow up was performed at 30, 60, and 90 days post surgery. Current arrangement is unnecessarily convoluted.
Line 91: before does not need to be capitalized
Line 91-105: repetitive – may be a download error?
Line 126: how was a loss of ROM less than 20o categorized? Unaltered? Seems inaccurate, perhaps preserved should be less than 20o reduction
Line 163: change have been to were
Line 163-170 can be summarized in a table and referred to – as written is a jumbled mess
Line 171 and onward: appropriateness of transcondylar screw size is better characterized by describing its size to the height isthmus of the condyle typically between 30 and 50% in most accounts. By weight really doesn’t well describe this relationship.
Line 175 onward – total number of screws proximal and distal to the fracture is not information that is helpful. It would be better to describe average/mean number of screws proximal and distal to the fracture per patient. Additionally plate length to bone length would also be helpful information as well as average/mean working length of constructs to better assess relative stiffness of constructs – were these typically long working lengths (screws towards the ends of plates only) with plates spanning greater than 75% of the bone length or were most holes filled with long plates and short working lengths. Or were shorter plates used with short working lengths. What was the screw density of your constructs.
Line 185: since this case (13) didn’t make it to the 30, 60, 90 day FU or 120 day FU the above sttements are not true – as these were not done for each dog – this dog die. Given this is a paper on outcomes I’m not certain this case fits the inclusion criteria. Something needs to be adjusted to keep this dog in the study.
Line 187 onward: if radiographic healing was detected were radiographs continued at 60 and 90 days post op? If not (I wouldn’t necessarily continue radiographing if considered healed) the description of follow up above needs to be adjusted – radiographs were taken at 30, 60, 90 days or until healing was detected/confirmed etc. Were they still rechecked (without radiographs once healing was detected (wording above would imply so – if not description needs to be changed to properly characterize the follow up once healing was confirmed 9ie did they just come back at 120 days if healing was detected at 30 days?).
Line 191 – says radiographs were taken at 120 days but the above description in methods is written such that radiographs is not clear in the term “patients were evaluated”
Figure 1 (line 196-200) – none of the distal screws appear in the images to be anything but perpendicular to the distal plate – they appear parallel to the screw proximal to them in both orthogonal images – would recommend annotating the images to show the divergence from perpendicular if that is possible or choose other radiographs that better demonstrate the divergence/angulation of screw placement.
Table 2 (line 202) in the pdf I received is the exact same table as Table 1. (the legned is different but the information in the table is the same).
Line 208: change has been to was
Line 209: change set to prescribed
Line 237: poorly worded “report apparently results”? needs to be reworded
Line 262: Sentence is a run on and needs to be reworded possibly broken up into 2 sentences. It is also missing a period at the end.
Line 265: multiple attempts at contouring is a known problem to avoid – not sure “special attention” is helpful for the reader to know what was done to avoid this – as I am sure most surgeon take special attention when contouring their plate with varying success.
Line 268-273: statement is confusing not sure what is attempting to be told here – there may be some missing punctuation – needs rewording and clarification
Line 273: needs to be a new paragraph I think.
Line 273-285: not sure where you are going with this controversy statement as no orthogonal plates were used and I know of no other highly successful techniques other than parallel plating for this fracture configuration in dogs. I am not sure what the controversy in human fracture repair has to do with veterinary medicine in this case – additionally, on the human side – people aren’t quadrupeds – so the necessary strength of repair is not similar. It doesn’t start to make sense until you get to the Garcia reference in line 280. Probably should begin with that information then discuss why you think your technique avoided having to do that. As written is convoluted.
Line 285: I haven’t used the PAX system but I do use locking plates – but I also haven’t ever had to resort to contouring the distal plate to be caudal. You haven’t shown in any objective way that you had to and were able to angle the screws away from the joint. This paper rests on this claim but no evidence is presented demonstrating this, as stated before the images don’t appear to have angled screws in the distal fracture segment (not saying they aren’t – just that this is not clearly demonstrated).
Line 286: the PAX system screw direction are still determined by plate design and position – just with small degrees of variability – 10 degrees – its not like you can angle the screw as much as one can with a cortical screw.
Line 288: the distal screws in your images are not bicortical – the rest of your bicortical screws in the constructs are similar to any other construct. I don’t think this statement is warranted as an advantage f this system unless other images can be provided
Line 291: change humeral to humerus
Line 291: this statement about how the polyaxial screw mechanism works should be in the earlier section describing the system – not buried in the discussion
Line 293: change peculiar to particular
Line 293 – pretty sure when inserted at an angle the polyaxial locking mechanism is weaker than when placed perpendicularly – or polyaxial systems by the nature of their design are weaker than the “traditional” locking systems (but still superior to cortical screw plate constructs) -and this should be stated as such here – as written it would appear the polyaxial locking is equivocal to other angle limited locking designs.
Line 321: needs rewording “resulted very useful” is not good grammar
Line 325: start new paragraph with discussion of ROM
Line 331: new paragraph with discsssion of bone healing
Line 339: new paragraph with discussion of infection
Line 340: poorly worded “requires generally its” poor grammar avoid use of “its” – is this word referring to complete removal of infection or implants – use of the pronoun its make it unclear – I’m sur eoyu mean the implants. Additionally poor wording for “because due to” needs re-wording as well
Line 341: “allowed to achieve” poor wording – whole sentence should be restructured
Line 345: avoid using the term progressive twice in the same sentence
Line 349: change “has been” to was
Line 354: was articular reduction to OA presented in the results section as a comparison? No statistical comparisons were made as such I would avoid this statement using the term correlate in this study – if this statement is only referring to the previous study cited then reword to be more clear.
Line 367: were short term outcomes of this study compared to previous? I don’t recall outcomes being compared – screw densities and plate placements were compared. Functional outcomes were just presented but not compared in the discussion (ROM, bone healing, infection, lameness – results were not compared to other studies in vet med – ROM was somewhat discussed in regards to human literature recommendations but not outcomes)
Author Response
Review Report 1
The author's presented a retrospective of 14 cases that used a specific implant design and report on the results of those interventions.
Please see comment below for suggestions to improve the manuscript's readability and usefulness for readers. Mainly I think more direct comparisons to veterinary literature in regards to their outcome to those of previous published results would be helpful and well as improved images depicting the polyaxial capability in use - possibly intraoperative image of the screw in the plate showing angled insertion or better radiographic images - or annotation of images to better demonstrate angle insertion of screws.
Thanks you for your suggestions and comments; we really appreciate your contribution and help to improve our work.
Line 46: change peculiar to particular Accepted and revised
Line 70: change are to were Accepted and revised
Line 77: wording awkward: possibly change to “the each patient underwent standard orthogonal radiographic projections of the humerus of the affected and contralateral limbs.” Accepted and revised
Line 84: change throught to through Accepted and revised
Line 83-86 – run on sentence – reword please Accepted and revised
Line 87 – why not just say radiographic follow up was performed at 30, 60, and 90 days post surgery. Current arrangement is unnecessarily convoluted. Thank you for the comment. Not in all cases we performed FU at 30-60 and 90 days. In some cases we detected bone healing at the first or second check. So in these cases as you can see in the table 1 (case number 1;2;4;6;7;8;9;10;11;12;14) we decide to avoid another sedation.
Line 91: before does not need to be capitalized Accepted and revised
Line 91-105: repetitive – may be a download error? Accepted and revised
Line 126: how was a loss of ROM less than 20o categorized? Unaltered? Seems inaccurate, perhaps preserved should be less than 20o reduction Thanks for the comment. We used the guidelines of reference 23
Line 163: change have been to were Accepted and revised
Line 163-170 can be summarized in a table and referred to – as written is a jumbled mess Thank you. Unfortunately in the previous submission there was an error in the table 2. Now All the results are included in the correct Table 2
Line 171 and onward: appropriateness of transcondylar screw size is better characterized by describing its size to the height isthmus of the condyle typically between 30 and 50% in most accounts. By weight really doesn’t well describe this relationship. Accepted and revised
Line 175 onward – total number of screws proximal and distal to the fracture is not information that is helpful. It would be better to describe average/mean number of screws proximal and distal to the fracture per patient. Additionally plate length to bone length would also be helpful information as well as average/mean working length of constructs to better assess relative stiffness of constructs – were these typically long working lengths (screws towards the ends of plates only) with plates spanning greater than 75% of the bone length or were most holes filled with long plates and short working lengths. Or were shorter plates used with short working lengths. What was the screw density of your constructs. Thank you for this comment. We believe that the evaluation of these biomechanical considerations is complex as it is a construct with double plate and the metaphyseal localization of the fracture does not always allow to have the plate equally distributed over the fracture. For these reasons we are not completely sure to give true information about biomechanic aspects of this particular fixation. Furthermore, our goal was to follow the principles for the internal fixation of joint fractures with rigid fixation and interfragmentary compression.
Line 185: since this case (13) didn’t make it to the 30, 60, 90 day FU or 120 day FU the above sttements are not true – as these were not done for each dog – this dog die. Given this is a paper on outcomes I’m not certain this case fits the inclusion criteria. Something needs to be adjusted to keep this dog in the study. Accepted and revised
Line 187 onward: if radiographic healing was detected were radiographs continued at 60 and 90 days post op? If not (I wouldn’t necessarily continue radiographing if considered healed) the description of follow up above needs to be adjusted – radiographs were taken at 30, 60, 90 days or until healing was detected/confirmed etc. Were they still rechecked (without radiographs once healing was detected (wording above would imply so – if not description needs to be changed to properly characterize the follow up once healing was confirmed 9ie did they just come back at 120 days if healing was detected at 30 days?). Thank you for this comment . Radiographs were taken until healing was detected, with a gap between one check to the other of 30 days. Some cases were healed at the first, some other at second; and some other at third radiographic check as reported in table 1. We confirmed that every dog was re-checked at 120 days post surgery except for the dog 13.
Line 191 – says radiographs were taken at 120 days but the above description in methods is written such that radiographs is not clear in the term “patients were evaluated” Accepted and revised
Figure 1 (line 196-200) – none of the distal screws appear in the images to be anything but perpendicular to the distal plate – they appear parallel to the screw proximal to them in both orthogonal images – would recommend annotating the images to show the divergence from perpendicular if that is possible or choose other radiographs that better demonstrate the divergence/angulation of screw placement. Ok thanks. we will add a better image that demonstrate inclination of the screws
Table 2 (line 202) in the pdf I received is the exact same table as Table 1. (the legned is different but the information in the table is the same). Accepted and revised
Line 208: change has been to was Accepted and revised
Line 209: change set to prescribed Accepted and revised
Line 237: poorly worded “report apparently results”? needs to be reworded Accepted and revised
Line 262: Sentence is a run on and needs to be reworded possibly broken up into 2 sentences. It is also missing a period at the end.Accepted and revised
Line 265: multiple attempts at contouring is a known problem to avoid – not sure “special attention” is helpful for the reader to know what was done to avoid this – as I am sure most surgeon take special attention when contouring their plate with varying success. Accepted and revised
Line 268-273: statement is confusing not sure what is attempting to be told here – there may be some missing punctuation – needs rewording and clarification Accepted and revised
Line 273: needs to be a new paragraph I think. Accepted and revised
Line 273-285: not sure where you are going with this controversy statement as no orthogonal plates were used and I know of no other highly successful techniques other than parallel plating for this fracture configuration in dogs. I am not sure what the controversy in human fracture repair has to do with veterinary medicine in this case – additionally, on the human side – people aren’t quadrupeds – so the necessary strength of repair is not similar. It doesn’t start to make sense until you get to the Garcia reference in line 280. Probably should begin with that information then discuss why you think your technique avoided having to do that. As written is convoluted Accepted .and revised
Line 285: I haven’t used the PAX system but I do use locking plates – but I also haven’t ever had to resort to contouring the distal plate to be caudal. You haven’t shown in any objective way that you had to and were able to angle the screws away from the joint. This paper rests on this claim but no evidence is presented demonstrating this, as stated before the images don’t appear to have angled screws in the distal fracture segment (not saying they aren’t – just that this is not clearly demonstrated). Thanks. Accepted and revised. New picture will be inserted.
Line 286: the PAX system screw direction are still determined by plate design and position – just with small degrees of variability – 10 degrees – its not like you can angle the screw as much as one can with a cortical screw. Thank you. Of course in this case you can angle the screw less than cortical screw but you can benefit about locking screw’s advantages
Line 288: the distal screws in your images are not bicortical – the rest of your bicortical screws in the constructs are similar to any other construct. I don’t think this statement is warranted as an advantage f this system unless other images can be provided We considered as bicortical screws the screws engaging both corticals of the epicondylar ridge in the distal segment without invading the supratloclear foramen.
Line 291: change humeral to humerus Accepted and revised
Line 291: this statement about how the polyaxial screw mechanism works should be in the earlier section describing the system – not buried in the discussion Accepted and revised
Line 293: change peculiar to particular Accepted and revised
Line 293 – pretty sure when inserted at an angle the polyaxial locking mechanism is weaker than when placed perpendicularly – or polyaxial systems by the nature of their design are weaker than the “traditional” locking systems (but still superior to cortical screw plate constructs) -and this should be stated as such here – as written it would appear the polyaxial locking is equivocal to other angle limited locking designs. Thank you for this comment . Accepted and revised
Line 321: needs rewording “resulted very useful” is not good grammar Accepted and revised
Line 325: start new paragraph with discussion of ROM Accepted and revised
Line 331: new paragraph with discsssion of bone healing Accepted and revised
Line 339: new paragraph with discussion of infection Accepted and revised
Line 340: poorly worded “requires generally its” poor grammar avoid use of “its” – is this word referring to complete removal of infection or implants – use of the pronoun its make it unclear – I’m sur eoyu mean the implants. Additionally poor wording for “because due to” needs re-wording as well Accepted and revised
Line 341: “allowed to achieve” poor wording – whole sentence should be restructured Accepted and revised
Line 345: avoid using the term progressive twice in the same sentence Accepted and revised
Line 349: change “has been” to was Accepted and revised
Line 354: was articular reduction to OA presented in the results section as a comparison? No statistical comparisons were made as such I would avoid this statement using the term correlate in this study – if this statement is only referring to the previous study cited then reword to be more clear. Thank you for this comment .This statement is referred to the cited articles
Line 367: were short term outcomes of this study compared to previous? I don’t recall outcomes being compared – screw densities and plate placements were compared. Functional outcomes were just presented but not compared in the discussion (ROM, bone healing, infection, lameness – results were not compared to other studies in vet med – ROM was somewhat discussed in regards to human literature recommendations but not outcomes) Thank you for this comment . We compared the general clinical outcome to previous studies in the conclusions
Reviewer 2 Report
The article deals with an interesting topic: to report the outcome of 14 dogs of Y-T 10 humeral fractures repair using paired Polyaxial Locking System (PAX) plates through a combined medial and lateral approach.
I suggest minor revision providing some comments:
line 83-86: please specify the bone healing assessment method in detail.
Lines 91-105: this section of the manuscript is repeated (lines 77-91) and should be deleted.
Line 108: please specify the osteoarthritic changes in detail.
Lines 118-130: like above this section is repeated fom 105-118: delete it.
Line 148: Please add what anti-inflammatory drugs were used.
Author Response
The article deals with an interesting topic: to report the outcome of 14 dogs of Y-T 10 humeral fractures repair using paired Polyaxial Locking System (PAX) plates through a combined medial and lateral approach.
Thanks you for your suggestions and comments; we really appreciate your contribution and help to improve our work.
I suggest minor revision providing some comments:
line 83-86: please specify the bone healing assessment method in detail. Thank you for this comment .We observed through x ray stage of union of the bone, the presence of callus formation and fracture line disappearance according to the reference (19) associated to the sentence.
Lines 91-105: this section of the manuscript is repeated (lines 77-91) and should be deleted. Accepted and revised
Line 108: please specify the osteoarthritic changes in detail. Accepted and revised
Lines 118-130: like above this section is repeated fom 105-118: delete it. Accepted and revised
Line 148: Please add what anti-inflammatory drugs were used. Accepted and revised
Reviewer 3 Report
The article deals with an interesting topic, T-Y fractures of the humerus, which mainly affect adults and are the result of torsional stress almost always based on trauma.
A first reading of the paper provides interesting information: a somewhat limited sample group, the use of new plates, and a postoperative outcome of considerable duration.
As a first suggestion to the authors, I suggest revising the keywords, selecting a more significant number (at least 5) as required by the guidelines for authors of the journal; moreover, they should not contain the exact words that already appear in the title.
There are several typographical errors:
line 30: a superscript appears in correspondence with the word region as if it were the result of copying and pasting.
line 91: capital letters appear in the middle of a sentence (i.e. before)
lines 117-118: punctuation might be revised entirely.
To the line 95: the acronym IOHC is often used: the definition of the pathology has recently changed to HIF because the pathogenetic hypothesis has changed. I would recommend using the current terminology. In the text it is reported that radiographic examination was used to analyze the predisposition to the intercondylar fissure. Recent literature highlights the limitations of radiographic examination and indicates the correct positioning to maximize the diagnostic potential of this technique. Does the author not believe that he is writing in the text the radiographic technique used and the limits connected with this technique?
Paragraph 3.1.: in English, the point, not the comma, should be used to indicate decimal places.
The article should be partially revised for the English language: there are considerable repetitions (see lines 65, 66...for each; 91 each patient, each patient), also the syntax might be revised (many sentences are not clear, and adverbs and subjects are not correctly placed). There are some grammatical errors, among the most evident at line 157, the present simple is used (goes) when the rest of the sentences are in the past tense; the discussion on line 237 states, "This article report".
The “materials and methods” section is difficult to understand in terms of content, presenting the same concept repeated several times (see paragraph 2.2: There are, at least in two places, entire paragraphs repeated twice).
At line 70, there is no explanation for minor and catastrophic complications: this section should be supplemented, obviously, with the bibliography attached.
Sentences at lines 87 to 90 could be reworded to clarify what was done at each control time.
Paragraph 2.3, despite being a surgical article, is not sufficiently detailed. The authors should give more detail regarding the surgery. Furthermore, leaving aside the different sizes of the scripts, it would be appropriate to add the anesthesiologic protocol applied, specifying if locoregional anesthesia was used.
Section 2.4 needs information on the NSAIDs used. The sentence on line 151 seems to contradict the ruling on the following line: I understand what the authors mean, but probably the concept needs to be reformulated.
In paragraph 3.2, line 211, bacteria culture should be specified, not simply culture.
Finally, the discussion should be better organized; there are many references to human medicine and references to veterinary literature are reduced to a few works. There are some typing and punctuation errors in this section.
Author Response
Thanks you for your suggestions and comments; we really appreciate your contribution and help to improve our work.
The article deals with an interesting topic, T-Y fractures of the humerus, which mainly affect adults and are the result of torsional stress almost always based on trauma.
A first reading of the paper provides interesting information: a somewhat limited sample group, the use of new plates, and a postoperative outcome of considerable duration.
As a first suggestion to the authors, I suggest revising the keywords, selecting a more significant number (at least 5) as required by the guidelines for authors of the journal; moreover, they should not contain the exact words that already appear in the title. Thank you for the suggestion. We improved the keywords.
There are several typographical errors:
line 30: a superscript appears in correspondence with the word region as if it were the result of copying and pasting. Accepted and revised
line 91: capital letters appear in the middle of a sentence (i.e. before) Accepted and revised
lines 117-118: punctuation might be revised entirely. Accepted and revised
To the line 95: the acronym IOHC is often used: the definition of the pathology has recently changed to HIF because the pathogenetic hypothesis has changed. I would recommend using the current terminology. In the text it is reported that radiographic examination was used to analyze the predisposition to the intercondylar fissure. Recent literature highlights the limitations of radiographic examination and indicates the correct positioning to maximize the diagnostic potential of this technique. Does the author not believe that he is writing in the text the radiographic technique used and the limits connected with this technique?
Thanks for the observation and we corrected with the new acronym HIF with updated bibliography. It’s a retrospective study that collected cases from 2016 to 2019 and at that time, we didn’t have a CT scan
Paragraph 3.1.: in English, the point, not the comma, should be used to indicate decimal places. Accepted and revised
The article should be partially revised for the English language: there are considerable repetitions (see lines 65, 66...for each; 91 each patient, each patient), also the syntax might be revised (many sentences are not clear, and adverbs and subjects are not correctly placed). There are some grammatical errors, among the most evident at line 157, the present simple is used (goes) when the rest of the sentences are in the past tense; the discussion on line 237 states, "This article report". Accepted and revised
The “materials and methods” section is difficult to understand in terms of content, presenting the same concept repeated several times (see paragraph 2.2: There are, at least in two places, entire paragraphs repeated twice). Accepted and revised
At line 70, there is no explanation for minor and catastrophic complications: this section should be supplemented, obviously, with the bibliography attached. Thank you for this comment .Accepted and revised
Sentences at lines 87 to 90 could be reworded to clarify what was done at each control time. Accepted and revised
Paragraph 2.3, despite being a surgical article, is not sufficiently detailed. The authors should give more detail regarding the surgery. Furthermore, leaving aside the different sizes of the scripts, it would be appropriate to add the anesthesiologic protocol applied, specifying if locoregional aneswthesia was used.
Accepted and revised. We confirm that we didn’t apply locoregional anesthesia but we used Fentanyl as intraoperative pain therapy.
Section 2.4 needs information on the NSAIDs used. Accepted and revised The sentence on line 151 seems to contradict the ruling on the following line: I understand what the authors mean, but probably the concept needs to be reformulated. Accepted and revised
In paragraph 3.2, line 211, bacteria culture should be specified, not simply culture. Accepted and revised
Finally, the discussion should be better organized; there are many references to human medicine and references to veterinary literature are reduced to a few works. There are some typing and punctuation errors in this section. Accepted and revised